# The Native Bees of Texas: Evaluating the Benefits of a Public Engagement Course

**DOI:** 10.3390/insects12080702

**Published:** 2021-08-05

**Authors:** Laurel Treviño Murphy, Shelly Engelman, John L. Neff, Shalene Jha

**Affiliations:** 1Outreach Program, Department of Integrative Biology, The University of Texas at Austin, 401 Biological Laboratories, 205 W 24th Street, Austin, TX 78712, USA; 2Research and Evaluation, Custom EduEval LLC, Austin, TX 78749, USA; engelman.shelly@gmail.com; 3Central Texas Melittological Institute, Austin, TX 78731, USA; jlnatctmi@yahoo.com; 4Department of Integrative Biology, The University of Texas at Austin, Austin, TX 78712, USA; sjha@austin.utexas.edu

**Keywords:** STEM outreach, learning outcomes, perceived knowledge, identification skills, pollinator insects, biodiversity, conservation

## Abstract

**Simple Summary:**

As concerns over bee population declines have entered the public consciousness worldwide, people are eager to learn about bees, the roles they play in our world, and how to conserve them. However, the public’s growing enthusiasm and efforts to conserve bees in North America are not always matched by their scientific knowledge of native bees. To satisfy a growing regional demand for knowledge about native bees, we have developed a public engagement program that aims to provide basic information about the native bees of Texas and their conservation guidelines based on science. At the University of Texas, Austin, we designed an outreach course with the objectives of teaching basic identification, diversity, ecology, and conservation of native bees and we implemented it on university botanical garden grounds. To gauge the course’s impact and quality, we integrated assessment tools into the course design. Evaluation results indicated that the course had a positive impact on participants who acquired specific topic knowledge and skills. The outreach course helped educate the public on native bees and benefitted participants, such as landowners and citizen scientists, who intended to apply their acquired knowledge and skills to specific conservation projects. It is relevant and timely to offer such courses, especially in regions that represent biodiversity hotspots for native bees and whose habitat is being fragmented and altered by rapid urbanization.

**Abstract:**

Declines in native bee communities due to forces of global change have become an increasing public concern. Despite this heightened interest, there are few publicly available courses on native bees, and little understanding of how participants might benefit from such courses. In October of 2018 and 2019, we taught the ‘Native Bees of Texas’ course to the public at The University of Texas at Austin Lady Bird Johnson Wildflower Center botanical gardens in an active learning environment with slide-based presentations, printed photo-illustrated resources, and direct insect observations. In this study, we evaluated course efficacy and learning outcomes with a pre/post-course test, a survey, and open-ended feedback, focused on quality improvement findings. Overall, participants’ test scores increased significantly, from 60% to 87% correct answers in 2018 and from 64% to 87% in 2019, with greater post-course differences in ecological knowledge than in identification skills. Post-course, the mean of participants’ bee knowledge self-ratings was 4.56 on a five-point scale. The mean of participants’ ratings of the degree to which they attained the course learning objectives was 4.43 on a five-point scale. Assessment results provided evidence that the course enriched participants’ knowledge of native bee ecology and conservation and gave participants a basic foundation in bee identification. This highlights the utility of systematic course evaluations in public engagement efforts related to biodiversity conservation.

## 1. Introduction

Insect diversity has profound impacts on terrestrial ecosystem functions that provide various benefits to humanity in the form of ecosystem services, such as pollination and nutrient cycling [1,2,3]. Most angiosperm species rely on animal-mediated pollination for their sexual reproduction [4,5,6], which supplies a quarter of mammal and bird diets with fruits and seeds [7,8]. It is estimated that insects provide agroecosystems across the globe with crucial pollination services for crops worth ~USD 500 billion annually [9], and that native pollinators, almost exclusively bees, pollinate fruits and vegetables worth ~USD 3 billion in the USA [3]. Bees are the most ecologically important pollinating insects contributing to most floral visitation events [10,11,12,13,14], followed by flies, butterflies, and moths [15]. Wild bees pollinate many crops effectively and significantly enhance fruit set, regardless of honey bee abundance [16], and studies have found positive relationships between wild bee diversity and fruit production [17,18,19], assumed to be driven by the functional complementarity of a diverse suite of pollinators in a community [18,20].

Native bee communities are particularly vulnerable to environmental stressors, which can alter species functionality and shift niches, leading to changes in community composition that impact ecosystem functions [20,21]. Environmental stressors include climate and land use changes, which are regional expressions of global change that have profound impacts on bee communities, including pollinator population declines, thereby affecting overall biodiversity levels [15,21,22]. Despite an increasing need to assess biodiversity, the taxonomic documentation of local flora and fauna has declined in the twentieth century [23], along with the support for insect taxonomists in academic and government organizations, even though more than 50% of terrestrial arthropod species remain undescribed amid a loss of biodiversity in human-altered landscapes [24]. The limited availability of taxonomists with species-level knowledge of bees impedes interdisciplinary studies of ecosystem biodiversity. Examples of these studies include characterizing native bee community composition, describing species distributions, and comparing the foraging behavior of native and non-native bees in the context of invasive plant species [24,25,26,27,28]. In particular, vulnerable native bee species should be surveyed with nondestructive methods, which involve visual species identifications. For example, nonlethal surveys have been employed for status assessments of restricted bumble bee species, pollination functions of native bees, and coarse assessments of community composition, such as relative abundance [29,30,31]. However, visual field-identification of bee species is restricted to researchers with taxonomic expertise, and the limited availability of taxonomists has led to bottlenecks for interdisciplinary studies in biodiversity and conservation science [23,24,26].

One way to overcome the taxonomic impediment in biodiversity conservation projects [23,24], is to implement a para-taxonomic approach in outreach initiatives. In para-taxonomy, experts group morphologically similar and closely related organisms into para-taxonomic units broader than species-level, termed morphospecies [28]. In biodiversity conservation studies, morphospecies have been used as a first step in sorting and identifying survey samples, to find patterns in taxonomic groups, and to describe gross species richness of single sites [28]. To teach identification skills, researchers have developed simple morphospecies identification guides that focus on few distinguishing morphological features, easily visible to the unaided eye [26,32,33,34,35,36]. Several citizen-science studies have demonstrated that experts can successfully teach “para-taxonomists” to identify morphospecies of large flying insects with simplified graphic ID-guides [26,32,34,35,36,37,38]. Indeed, experts around the globe have used para-taxonomy to train volunteers to identify various organisms, and species diversity has been documented for plants, birds, and large insects with well-established citizen science protocols, forming the backbone of community and citizen science [32,34,35,37,38,39,40,41]. In expert-assisted community science programs, morphospecies ID-guides can be used in a stepwise progression from para-taxonomic/genera identifications performed by citizen scientists, to taxonomic/species identifications performed by experts [41]. However, bee photo ID-guides can pose limitations even for experts and require very careful design [42].

Building on the success of training non-experts to identify morphospecies, experts have further taught volunteers to monitor wildlife species populations with simplified survey protocols that collect para-taxonomic data [30,32,34,35,37,41]. Using simple survey protocols, volunteers can perform visual identifications of flower-visiting insects to document incidental observations or conduct basic pollinator surveys [32]. For example, experts have taught volunteers to conduct basic surveys of native bee groups in California, spawning similar pollinator monitoring projects across the United States [30,32,35,37]. Other applications include monitoring the status of regional bumble bee populations [24]. While expert-curated data are essential to estimate community parameters, such as species composition and distribution, which inform species conservation decisions [26,27,28], volunteers trained by experts to identify bees at similar morphospecies levels can collect para-taxonomic data with accuracy levels sometimes comparable to that of academic researchers [35,37]. For example, supervised volunteers have helped researchers obtain para-taxonomic data used for coarse assessments of community parameters, such as community composition, proportional abundance, and species dominance of native bee functional groups and large bee species [31,37,41]. Some species monitoring protocols have also collected data to determine ecosystem services, such as pollination functions of native bees (e.g., Native Bee Watch, Bee Watch, Great Pollinator Project, Great Sunflower Project) [35,40,43,44,45] and butterfly and bee habitat quality indices [34,35]. The reliability of para-taxonomic data obtained in pollinator surveys depends, to some degree, on the observer’s identification skills, which improve with expert-supervised field training and practice [32,35,37,38].

Modified survey protocols can also be used for purposes other than community science projects. For example, landowners or managers can gauge the impact that management practices have on pollinator habitat by identifying native bees and monitoring their population trends, provided they have reliable para-taxonomic data that inform sound wildlife management plans [34,38,39,46]. These survey methods offer a low-cost, practical, time-saving approach to monitoring bee populations across regions where expert involvement is not available [9,30,38], and standardized survey protocols can provide comparable data across research or conservation projects [47,48,49]. Additionally, nondestructive survey methods, which use visual identifications, pose low risks for vulnerable bee species populations [29,31].

In many public engagement programs, experts have focused primarily on training volunteers to identify organisms for monitoring purposes, while in other programs, experts have taught community members about native bee monitoring in the context of their importance as pollinators in natural landscapes [36,37]. Outreach courses that teach participants native bee identification in the context of biodiversity may lead to a better understanding of ecosystem functions and a greater appreciation of the ecosystem services that native bee communities provide. Though an ecological context has not typically been incorporated into outreach courses that focus on native bees, it is an important educational approach, given the public’s increased interest and support of bee conservation, which currently exceeds people’s understanding of bee diversity and ecology [50]. Additionally, this educational approach may help cultivate participants’ attachments to nature while instilling a civic ecology component to public engagement initiatives [40,51]. Equally important, though not often prioritized in public engagement work, evaluative efforts are critical to ascertain the impact of outreach courses on their participants and communities [36].

We designed and implemented the “Native Bees of Texas,” public course to teach identification skills in the context of native bee diversity, ecology, and conservation, as part of an outreach program at The University of Texas at Austin, Integrative Biology Department. It was especially relevant to teach this course in the understudied region of the south-central U.S., an area with high bee diversity characteristic of xeric ecosystems in North America [13,52]. In fact, the twelve ecoregions spanning Texas [53,54] comprise a biodiversity hotspot with approximately 1100 of the 4000 native bee species in North America [13,55] (John Neff, pers. comm.). In addition to its ecological diversity, Texas was the nation’s second largest producer of some agricultural goods in 2011 and the fourth in terms of crop cash receipts in 2019, including many pollinator-dependent crops, such as cotton, melons, beans, and sunflower seeds [56]. It is also one of the most rapidly urbanizing states, with three of the eight fastest growing U.S. cities [57], and is likely to experience more frequent extreme climate events in the future [58], where drivers of global change will put both natural and human-dominated landscapes at risk. This is especially critical for bees, whose populations tend to decline with land use changes that decrease floral resource levels and who have a higher negative response to anthropogenic disturbance compared to other pollinators [15]. Given this combination of factors, we posit that native bee conservation in Texas warrants our urgent attention, and that expanding science literacy is key to advancing this objective [33]. An outreach course focused on native bees is relevant and timely in this region, where expert-trained volunteers could help document and monitor bee diversity with the use of participatory approaches, as in previous studies [30,34,35,37,41,49].

In this paper, we present findings from the “Native Bees of Texas” course, which we conducted in the botanical gardens and teaching facilities of The University of Texas at Austin Lady Bird Johnson Wildflower Center, in October of 2018 and 2019. The goal was to teach participants the basic principles of native bee identification and ecology and to provide guidelines for the management and conservation of local pollinator habitat. To evaluate the course for quality improvement, we designed and implemented a combination of assessment instruments that answer the following questions: (1) what were participants actual gains in ecological knowledge and identification skills?, (2) what were their perceived gains in ecological knowledge and identification skills?, (3) how did participants perform in field identifications relative to experts?, (4) how confident and motivated were participants?, (5) how did participants plan to apply their acquired knowledge and skills?, and (6) what steps should be considered for future courses?

## 2. Methods

### 2.1. Course Objectives

We designed the “Native Bees of Texas” course to give participants an overview of life history and ecology, diversity and identification, and habitat conservation. To increase student engagement and learning, the course design incorporated opportunities for active-learning, known to improve learning outcomes [59,60,61,62], and place-based learning, known to increase positive outcomes [63,64]. Course activities included observing insects both indoors and outdoors in natural settings where the instructor encouraged participation during presentations. To evaluate the course’s overall impact and efficacy, we designed a combination of diagnostic and summative tools that assessed participants’ learning outcomes and perceptions, as well as the applicability of course content to participants’ lives [65]. Similar assessment instruments have been implemented in other outreach activities focusing on native bees [36]. The assessment instruments we used for course quality improvement are detailed in the Course Evaluation section below.

### 2.2. Course Format & Composition

The half-day course format expanded on previous bee courses [33,35,36,37,38,41,66], in which instructor presentations on bee ecology, diversity, and identification were followed by indoor or outdoor activities that used simple insect identification guides (see Supplementary Scheme S1). Course content was organized in three thematic sections related to life history and ecology, identification and diversity, and habitat conservation. The bee life history-ecology section focused on diet, foraging behavior, and nesting and sociality behavioral traits that tie into pollination functions across ecosystems. The bee identification-diversity section introduced participants to the diversity within the six Apoidea families in North America, following Michener’s (2000) taxonomic classification. This section focused on identifying bee morphospecies by behavioral and morphological features that are visible to the unaided eye. Identification was taught in two indoor (lab) activities at progressively finer taxonomic levels of resolution, as in Ullman et al. [32], in three steps: first by distinguishing bees from similar flower-visiting insects, second by distinguishing native bees from western honey bees, and third by identifying major para-taxonomic groups that roughly coincide with the six bee families native to North America and Texas (Supplementary Scheme S2A,B). Material for teaching bee identification skills included photo-illustrated species lists of local garden bee fauna and the “Bees of Central Texas Guide” (Supplementary Scheme S2C–E), similar to the simple identification keys designed for the public by previous researchers [30,35,36,37,38,39]. We based the native bee para-taxonomic groups used in the course ID-guide on biological knowledge and taxonomic criteria gleaned from peer-reviewed literature, and the authors’ taxonomy, ecology, and biology experience. Finally, the pollinator habitat conservation section focused on “Management Recommendations for Native Insect Pollinators in Texas” [46] and drew on participants acquired knowledge of native bees’ diet, life histories, and behavior, applied to pollinator habitat management (Supplementary Scheme S2F,G). This section recommended gardening and landscaping practices to maintain native bee habitat and introduced a list of common native prairie plants visited by native bees in Central Texas (Supplementary Scheme S2H,I).

In the two indoor labs, participants examined pinned insect specimens with stereo microscopes and loupes, using an instruction sheet on comparative morphology and the native bee ID guide (Supplementary Scheme S3A,B). In the first lab, comparisons between flies (Order Diptera), and wasps and bees (Order Hymenoptera), were based on seven easily visible morphological features that distinguish dipterans from hymenopterans: the body regions, eye location, wings, legs, antennae, branched hair, and coloration [32,33]. In the second lab, participants learned to distinguish native bees from non-native honey bees (*Apis mellifera*) (Apoidea superfamily), and they were introduced to native bee diversity with examples from five of the six North American bee families found in Texas (Supplementary Scheme S4). As in previous studies, the focus was on four easily visible morphological features characteristic to all bees: body form, size, hair location, and coloration [32,33]. The two additional, non-visual, features listed in the bee ID-guide, nesting sociality and pollen preference, had previously been discussed in the presentations. Pollen preference (lecty) was explained by showing examples of oligolectic and monolectic bee species that restrict their pollen collection to a single plant family, genus, or species, in contrast to polylectic generalists.

The third activity consisted of a 30-min observation session of live flower-visiting insects in the native plant gardens on-site, where participants could use the native bee ID guide and local species list to identify bees on the wing. During this activity, participants could observe the morphological and behavioral features of bees that had been discussed in presentations and were described in the ID-guide: buzzing flight, nectar/pollen gathering, and floral fidelity. In the 2019 class, we also conducted paired insect surveys as in [32,34,35,37]. Per the standardized sampling protocol outlined in Kremen et al. [37], each team was composed of two students, working in observer-recorder pairs, and one expert. Three teams each performed one 15 min survey along the same 80 m × 2 m transect, where a student pair and one expert in the same team observed insects while walking from opposite ends of the path and crossing midpoint (Supplementary Scheme S5A). While students identified bees by their common name at a para-taxonomic group level, experts identified bees by scientific name at a genus or species level. Participants recorded their identified insect counts on standardized data sheets, which the instructor tallied for each of the three teams by equating students’ para-taxonomic groupings to experts’ genera (Supplementary Scheme S5B).

Additionally, to link bee life history traits with pollination functions, the following basic concepts were highlighted in presentations, printed material, and activities. Pollinators and flowering plants co-evolved and bees’ ephemeral life spans were often in synch with flowering plant phenology. Bee characteristics include collecting and carrying pollen loads on patches of branched hair, females provisioning nests with pollen/nectar, and exhibiting floral fidelity (the tendency to consecutively visit flowers of the same plant species on a foraging trip). After course presentations and activities, students were asked to characterize the bees in Texas, and the instructor summarized that, most native bee species in North America are solitary, ground-nesting, generalists (see Supplementary Scheme S6A–D).

### 2.3. Course Evaluation

We used a combination of seven tools, explained below, to assess Performance-based Knowledge, Perceived Knowledge, Overall Course Ratings, and Open-ended Feedback. We also collected background information with anonymous forms to obtain participant Demographics. Participants’ voluntarily completed survey forms to evaluate the course and instruction (see Supplementary Schemes S7–S9). Specifically, we conducted the following:To assess Performance-based Knowledge gains in both years, we used the correct responses from a 10-question multiple-choice test to compare a respondent’s individual pre-course score to their post-course score as in [67], which provided an indicator of the knowledge gained during the course as in [27]. Test questions had one correct answer out of 5 multiple choices, and test scores were reported as percent correct responses. Participants responded to 10 questions, the first five of which pertained to Visual Identification Skills and the latter five of which pertained to Ecological Knowledge specific to native bees. For the skills assessment section, participants were asked to visually identify insects in five photos (including honey bees, native bees, and bee-mimic flies and wasps); these same photo identifications were conducted both before and after the course [33]. The ecological knowledge section focused on bee characteristics, native bee nesting and foraging behavior, ecosystem functions, pollination services, and threats to populations. Both knowledge and skills were assessed pre- and post-course [33,36] (Supplementary Schemes S7 and S8).To further assess Performance-based Knowledge gains in 2019, expert and student volunteers performed standardized paired surveys of flower-visiting insects (Insect Surveys). We checked participant’s bee identification accuracy by comparing the two data sets, checking student’s observations with expert’s insect identifications [35,37] (Supplementary Scheme S5A,B).To assess General Perceived Knowledge gains in 2019, participants self-rated their pre- and post-course overall knowledge of bees in a retrospective manner using the statements “Before participating in this workshop…”, and “Now, I would rate my knowledge of native bees as…”. We used a categorical approach of rating on a 5-point Semantic Differential Scale as in [68] with the options: “poor, fair, good, very good, excellent” as in [36,65] (Supplementary Scheme S9).To assess Specific Perceived Knowledge gains in both years, participants retrospectively rated their knowledge across six topics using the statement “As a result of today’s workshop, I am better able to…” These six learning objective topics referred to their knowledge of the importance of bees in ecosystems, distinguishing bees from other flower-visiting insects, using a basic ID guide to identify native bees, using taxonomy to learn more about bee diversity, and knowing which native plants are most beneficial for native bees. Participants used a 5-point Likert scale to rate their knowledge using the terms “strongly disagree, disagree, neutral, agree, strongly agree” [69].To obtain Overall Course Ratings in both years, using three criteria, informative, useful, and engaging, participants used a 5-point scale ranging from (1) “not”, to (5) “very”. Specifically, “informative” ratings indicated concordance between course content level and participants’ base versus acquired knowledge levels, “useful” ratings indicated its relevance and applicability to participants lives, and “engagement” level indicated how active the learning environment was for them.Finally, in Open-ended Feedback obtained in both years, we gathered participant’s intended Applications of acquired knowledge, course Highlights, and suggested Improvements. Participants’ responses to open-ended questions indicated their overall experience of the course, their changes in attitude toward science subjects, and their confidence and motivation levels post-course.To obtain Demographic Information in both years pertaining to age, gender, race, income, education, occupation, and residence, participants were asked to voluntarily fill an anonymous/confidential background information form (see Appendix A).

### 2.4. Data Analysis

In both years, we administered the same test with ten multiple-choice questions to each participant before and after the course. We analyzed the percent of correct responses on the pre/post tests for each participant and conducted a Mann–Whitney U test as in [70] to assess statistically significant differences between correct pre- and post-course test responses as in [65,67]. To be conservative, we did not pool test results from the two years due to variations in the proportion of respondents, participant backgrounds, instructor’s experience, and wording on assessment forms. Furthermore, we added two assessment tools in the 2019 course: the retrospective general perceived knowledge rating, and the insect survey. To analyze the 2019 Insect Survey results, we first established comparable taxonomic categories where bee groups identified by students correspond to bee genera identified by experts, [35,37]. We subsequently tallied observations of bee genera for student and expert data and grouped both data sets into three categories: all flower-visiting insects, all bees (non-native honey bees and native bees) and only native bees. For both student and expert insect survey data, we calculated abundance means and proportions for butterflies, flies, wasps, and bees, but we did not statistically compare the means due to a low sample size (N = 3). We analyzed the categorical data from participants’ 2019 retrospective ratings of their General Perceived Knowledge and used a Mann–Whitney U test to assess statistically significant differences between pre- and post-course responses as in [70]. No statistical analyses were done on the non-paired data from the ratings of Specific Perceived Knowledge of six topics, the Overall Course Ratings, or the Demographics, though categorical data was converted to percentages for visualization. For the Open-ended Feedback, we grouped participants’ answers into response categories that we established during the data analysis, then we quantified the proportions of participants that contributed comments in each category.

## 3. Results

### 3.1. Course Attendance and Demographics

A total of 20 participants attended the 2018 course and 21 attended in 2019. Only participants who completed both pre- and post-course surveys were included in the analysis; therefore, we evaluated 12 participants in 2018 and 20 participants in 2019. There was a 61% response rate from the total number of participants of both years. Most participants were white, college-educated, working women, over 55, with a range of backgrounds, including education, health care, IT, real estate, environmental education, engineering, farming, and ranching. Most participants additionally self-identified as gardeners or naturalists (Appendix A).

### 3.2. Evaluation for Course Quality Improvement

#### 3.2.1. Performance-Based Knowledge Gains: Test

Participants’ gained performance-based knowledge and skills across both years. The overall mean of correct responses on the post-course test rose significantly in both years, by 27.2% (±4.90) in 2018 (*p* = 0.009) and by 23.5% (±7.72) in 2019 (*p* = 0.000) (Figure 1A,B). The differences between the overall means of the five topics related to the Visual Identification Skill section were 25.6% (±7.93) in 2018 and 13.6% (±10.05) in 2019, with a mean improvement of 19.6% and correct responses between 80% and 82% post-course across both years. The mean differences for the five topics related to the Ecological Knowledge section were 28.8% (±6.60) in 2018 and 33.4% (±10.82) in 2019, with a mean of 31% and correct responses at 93% (±0.05) or more post-course across both years (Appendix A).

Within topics of Visual Identification Skills, significant gains were seen in respondents’ ability to distinguish the sex of a native bee, which rose 46% in 2018 (*p* = 0.028) and 42% in 2019 (*p* = 0.007). Participants’ ability to distinguish non-native honey bees (*Apis mellifera*) from native bees, although not significantly different, was 27% higher in 2018 (*p* = 0.193) and 26% higher in 2019 (*p* = 0.056); followed by their ability to distinguish house flies and bombyliid flies from bees, which was 37% higher in 2018 (*p* = 0.102) and 16% higher in 2019 (*p* = 0.331). The ability to discern hover flies from bees rose 18% in 2018 (*p* = 0.303) and 0% in 2019 (*p* = 1.000), while no gains were made in the ability to discern bee mimic wasps and cuckoo bees, 0% in 2018 (*p* = 1.000) and −16% in 2019 (*p* = 0.187). Overall, participants made significant gains in sexing bees and in their conceptual knowledge of characteristic bee features. Combined test results showed that participants’ visual identification skills enabled them to correctly identify at least 80% of hover flies, wasps, and bees.

Within topics of Ecological Knowledge, significant gains were seen in respondents’ knowledge of Nesting Behavior, which rose 36% both years (*p* = 0.035 in 2018 and *p* = 0.005 in 2019), knowledge of Pollination Services, which rose by 45% in 2018 (*p* = 0.015), knowledge of Ecosystem Functions, which rose by 26% in 2019 (*p* = 0.021), and conceptual knowledge of characteristic Bee Features, which rose by 74% in 2019 (*p* < 0.001) and by 36% in 2018 (*p* = 0.068). Participants knowledge in the topic of bee population declines was not significantly different pre- and post-course; 9% in 2018 (*p* = 0.363) and 15% in 2019 (*p* = 0.083) (Figure 1A,B) (see Appendix A).

#### 3.2.2. Performance-Based Knowledge Gains: Insect Survey

During the paired insect survey, volunteers observed non-native honey bees (*Apis mellifera*) and native bees, skipper butterflies, bombyliid flies, and Mexican honey wasps among the flower-visiting insect community. In both the expert and student data sets, the proportional abundance of insect groups followed similar trends (Figure 2A). Of the 161 bees that experts observed, native bees represented a 0.61 proportion of the community, while honey bees represented a 0.25 proportion of the community. Of the 98 bees that students observed, native bees represented a proportion of 0.58 of the community while honey bees represented a 0.22 proportion. Experts identified eight native bee genera with the most abundant, a 0.26 proportion, being leafcutter bees (*Megachile* sp.), large carpenter bees (*Xylocopa* sp.) (0.16), hairy leg (*Centris* sp.) or longhorn bees (*Melissodes* sp.) (0.14), and bumble bees (*Bombus* sp.) (0.14), while the least abundant genera were tiny dark bees (*Lasioglossum* sp. and *Ceratina* sp.) (0.01) and cuckoo bees (*Coelioxys* sp.) (0.01). Students reported seven native bee groups, five of which correspond to the experts’ identifications, and included bumble bees (0.44), hairy-belly leafcutter bees (0.16), large carpenter bees (0.06), longhorn bees (0.03), and tiny dark bees (0.01). Within just the bee community, honey bee proportional abundance was approximately a third of all native bee groups for both the expert and student data (Figure 2B) (Appendix A).

#### 3.2.3. General Perceived Knowledge Gains

For the general perceived knowledge ratings, there was a significant increase from the pre-course mean of 1.65 (±0.21) to the post-course mean of 3.35 (±0.18) (*p* < 0.001) on a five-point scale. Specifically, 80% of participants perceived their pre-course knowledge as poor or fair, while only 10% perceived their post-course knowledge as poor to fair. Similarly, while 5% of participants perceived their pre-course knowledge as very good or excellent, 35% perceived their post-course knowledge as very good or excellent (Figure 3) (Appendix A). 

#### 3.2.4. Specific Perceived Knowledge Gains

For the specific perceived knowledge ratings of six topics (learning objectives for participants), the mean exceeded 4 on a five-point Likert scale (1—strongly disagree, 2—disagree, 3—neutral, 4—agree, 5—strongly agree). In both years, students perceived their greatest knowledge in bee ecology and conservation followed by bee identification. Specifically, students gave the highest rating for “Understanding the importance of native bees in ecosystems” (2018 mean 4.75, 2019 mean 4.58), followed by “Using best management practices to conserve native bee habitat” (2018 mean 4.50, 2019 mean 4.74). Students assigned the next highest ratings for “Distinguishing native bees from similar flower visitors” (2018 mean 4.50, 2019 mean 4.26), followed by “Knowing which Texas prairie plants to use for native bee gardens” (2018 mean 4.25, 2019 mean 4.53). Students assigned the lowest rankings to “Using taxonomy foundations to learn about native bee diversity” (2018 mean 4.33, 2019 mean 4.37) and “Identifying common Texas native bees with course guides” (2018 mean 4.33, 2019 mean 4.00) (Figure 4) (see Appendix A).

#### 3.2.5. Overall Course Ratings

Participants assigned overall course ratings based on three criteria: informative, useful, and engaging. Taking the mean of these three criteria, both years were rated above 4.6 on a five-point scale by 89.5% of participants in both courses. In 2018, participants gave equally high ratings for all course characteristics, Informative (5.0), Useful (5.0), and Engaging (5.0). In 2019, participants gave the highest rating for Informative (4.95), followed by Useful (4.74), and Engaging (4.26). Evaluating the course ratings across both years, 97.5% of participants rated the course as “very informative,” 89.5% rated their acquired knowledge and skills as “very useful,” and 71% rated the course as “very engaging” (see Appendix A).

#### 3.2.6. Open-Ended Feedback

In the Open-ended feedback, participants described two primary application areas for their acquired knowledge and skills: (1) gardening/land management and (2) educational projects. Approximately 91% of participants in 2018 and 74% of participants in 2019, 82% over both years, indicated they intended to apply their gained knowledge to home gardening or land management on farms or ranches. Overall, participants stated they planned to improve habitat to increase native bee diversity in their gardens by leaving bare ground and reducing mulch for ground-nesting bees, eliminating insecticides, converting lawns to pollinator gardens, and enhancing prairies with native wildflowers and grasses to increase the abundance and diversity of bees. Specifically, 9% of participants in 2018 and 10.5% in 2019 described the importance of improving native bee habitat for pollinator conservation, and 27% of participants in 2018 and 10.5% of participants in 2019 stated that knowledge of native bee nesting would influence their gardening and land management practices. The same percentages in both years expressed the importance of learning about native bee diversity and identification. Overall, 27% of participants in 2018 and 26% in 2019, 26.5% over both years, intended on applying their knowledge about native bees to educational projects by sharing it in master gardener clubs or through educational material, such as the development of photographic ID guides. Most participants in 2018 and half in 2019 appreciated the expert instruction, slide-based presentation style, and direct experience observing bees in lab and field activities. Overall, 42% further stated they valued the course format and expert instructor’s presentation style with open-ended questions that invited participation. Further, 18% of participants in 2018 and 10.5% in 2019 (14.25% across both years), found the printed material and slides to be helpful resources. Overall, 28% of the participants, indicated that they most enjoyed the experience of directly observing flower-visiting insects in both the labs and the field, which complemented their learning process about bee diversity and identification. Among future improvements, participants identified course duration/pace and instruction. In both 2018 and 2019, 21% of participants suggested a longer course with a slower pace that would allow more time to spend in the gardens and share information among participants. Specifically, in 2018, 9% of participants suggested including more native plant identification activities, while in 2019, 5% of participants suggested focusing more on insect identification activities (see Appendix A, Applications of Acquired Knowledge; Appendix A, Effective Course Aspects; Appendix A, Suggested Course Improvements).

## 4. Discussion

In this study, we evaluated the overall impact and efficacy of a native bee outreach course and presented findings from course quality improvement assessments based on participant feedback and learning outcomes. Good response rates from course participants, who voluntarily completed surveys and tests, allowed for a robust evaluation of the course’s impact and efficacy [71]. We documented significant gains in the overall test score means, large pre/post-course differences in participants’ knowledge of native bee ecology and identification skills, and greater confidence and motivation in participants who intended to conserve native pollinator habitat. We also found concordance between participants’ perceptions of their knowledge levels and their test performances. These results are encouraging, given that, despite an increase in the public’s interest in native bee conservation, we have found few publicly available science courses on this topic that employed diagnostic assessments of acquired knowledge and skills [33,36,38]. To our knowledge, this is the first formally implemented and evaluated native bee identification and ecology course in the south-central US.

First, we documented notable post-course gains in participants’ bee identification skills. Overall, we found large but not significant increases in participants’ ID skills, likely due to high heterogeneity in the data set. However, participants in 2019 had significant gains in their conceptual knowledge of characteristic bee features, and across both years, their visual skills enabled them to identify most (~80%) hover flies, wasps, and bees. Future courses should focus on the direct observation of morphological differences between bees and wasps. Past studies employing similar methods reported similar results for post-training assessments [33,36,38]. One study documented that ~85% of hover flies, wasps, and bees were identified by citizen scientists [38]. A second study documented that ~72% of native bees were identified by agricultural workers [33]. A third study documented that ~96% of participants increased their ability to identify native bees [36]. Two additional studies focused on assessing post-course identification skills using the same student-expert insect survey protocol and documented similarities in the paired data sets [35,37]. As in the previous studies, we documented similar proportional abundance of flower-visiting insect groups, including a high proportion of native bees*,* in the student-expert paired data sets [35,37]. Specifically, in our study, the total number of bee genera observed were similar (student: 7, expert: 8), the most abundant native bees were large-bodied, and the most abundant genera were the same: hairy-belly leafcutter bees (*Megachile* sp.) and bumble bees (*Bombus* sp.). Although the sample size in this pilot study was too small to draw conclusions, in general, if results of para-taxonomic sorting show clear and biologically meaningful patterns, the sorting is likely to be reliable [28]. Future studies using the same methodology with a larger sample size, would permit a more robust analysis. Synthesizing across assessment results, this study resonates with past work that indicates a strong capacity for non-scientists to learn to identify bee para-taxonomic groups when trained by experts using photo-based presentations, photo-illustrated ID-guides, and direct observation of insects.

Second, we documented post-course gains in participants’ ecological knowledge of bees. Most participants had higher post-course knowledge levels across both years with significant gains in their understanding of bee life history in both years, ecosystem functions in 2019, and pollination services in 2018. These knowledge gains speak to the effectiveness of framing bee identification courses in an ecological context. Two other studies also assessed participants’ general ecological knowledge of bees before and after presentations by experts. One urban community outreach workshop that focused on the importance of conserving native bees in native landscapes, documented post-activity knowledge gains for most participants [36], while a rural field workshop that focused on bee diversity, wild bee pollination and honey production, and bee population declines, documented post-activity gains in knowledge, awareness, and perception for most participants [33]. While we documented similar gains in four ecological topics, we also note differences in methodology and participant background. Where we facilitated structured classroom presentations and activities for college-educated, urban/suburban dwellers who had little background in native bees, one study involved interactive field presentations and activities for a North Carolina public with disparate ages and backgrounds [36], and the other study involved a brief field presentation for adult rural agricultural workers who had a primary-level education and a deep cultural knowledge of beekeeping in India [33]. Despite differences in methodology and demographics, these studies show that non-scientists of very different backgrounds can gain substantial knowledge about the ecological roles of bees through interactive short courses.

Third, we documented gains in participants’ confidence, motivation, and positive attitudes toward science, as well as a high specificity in how they intended to apply their knowledge. Evaluating the General Course Ratings across both years, most participants rated their acquired knowledge and skills as “very useful”, and similarly high proportions expressed confidence in applying their knowledge to conserve native bees in their own communities. Interestingly, participants expressed more confidence in their knowledge of bee ecology than in their bee identification skills, reflecting concordance between their perception and their performance. From the Open-ended Feedback, it was clear that most participants intended to apply their knowledge to improve pollinator habitat for greater native bee diversity and abundance. Participants further indicated how their perspective on land management had changed after learning about native bee diversity, solitary bee nesting, and foraging behavior, and that they intended to change their management practices accordingly, by adding native plants for native bees, reducing the use of mulch to leave more bare ground for ground-nesting bees, and avoiding the use of insecticides on their land. Participants in previous studies were also more willing to add ‘bee-friendly’ plants to their gardens after gaining greater understanding through educational activities [36,40]. Participants in our course highlighted that the most enjoyable learning experiences were observing flower-visiting insects in their native habitat. This feedback resonates with past field courses where participants indicated that outdoor learning experiences were effective and had positive impacts on their perception of the importance of bee habitat [36,40]. Participants also expressed they highly valued the course’s participatory style, a pedagogical approach that has been shown to support both critical thinking and active learning while increasing learning outcomes [60,61]. While the course motivated most participants to conserve and document native bee diversity on their land or in their neighborhood, some participants were further inspired to educate others in their communities about native bees. This feedback reflects a spirit of civic ecology that goes beyond participatory valuations of biodiversity or ecosystem services to engage people in activities that improve the environmental quality of their communities and increase their well-being [51].

In the open-ended feedback, participants also revealed a potential for stronger connections between natural science educators and landowners who want to become better land stewards of pollinator habitat. It is important that these outreach efforts reach private landowners in Texas where approximately 95% of open space is in private holdings [72]. Much of the 4.1 million acres in private agricultural, ranching, or timber operations have been converted to other land uses [73] or sell as hunting land [74], which may be appraised under the state’s tax code for wildlife management [75]. Under the state’s wildlife program, tax valuations may be granted to rural landowners who implement land management practices to sustain indigenous wildlife populations [75,76]. The program has helped rural landowners, who manage the public’s wildlife, to retain their costly land [77] by reducing the tax burden from rising land values [78]. In 20 years, this beneficial program has spread to 4.7 million acres of privately owned land, eight times the acreage of state natural areas [79], which may have slowed the fragmentation of open space, often triggered by economic imbalances and urban sprawl that cause a rural exodus [72]. Recently, Texas Parks and Wildlife Department extension biologists proposed including insect pollinators in the wildlife program and, in collaboration with academics and NGOs, have developed habitat management guidelines for native bee habitat that include pollinator surveys to sustain native bee communities [46]. Rural landowners may be further incentivized to conserve habitat if they knew the wildlife program applies to pollinators. Increased awareness of local pollinators is tremendously valuable given the ecosystem services that insects provide, mostly from native bee pollination, which have been estimated in the tens of billions of dollars [3,8,9]. This represents economic gains that could potentially help offset the costs of the tax-discount wildlife program. Texas lands within the wildlife program were estimated to be worth, on the market, USD 10.3 billion, and the discount for wildlife management amounted to a significant percentage of that value [79]. To help protect the public stake and support extension biologists who develop wildlife management plans in the wildlife program [72,80], paired courses on native bees and pollinator monitoring could be tailored to increase stakeholders’ abilities to manage habitat. This could add momentum and effectiveness to conservation initiatives. Overall, course participants’ positive feedback related to land stewardship, highlighted the great potential to promote pollinator conservation among landowners, especially given that some course participants managed ranches or farms.

Future directions and improvements for outreach courses were also identified in the open-ended feedback. A quarter of the participants suggested a longer course or series of workshops with more time to explore the gardens and share information among themselves, strengthening previous study proposals to increase outdoor team activities that can facilitate better learning outcomes [62,63,64]. This may be feasible if two instructors team-taught labs, guided outdoor insect observations, and facilitated small group learning activities and discussions, all of which can have great effects on active learning in STEM disciplines [60,62]. Additional course improvements include diversifying homogenous participant demographics. In our study, participants were mostly white, college-educated, working, or retired adults that learned of the event through the websites of the University of Texas College of Natural Sciences, Lady Bird Johnson Wildflower Center, and Texas Master Naturalist/Gardener chapters. This challenge could be partially overcome by including more diverse advertisement channels and offering a free course in an urban location that is accessible by public transportation and would likely attract residents from a more diverse demographic. To attract a more diverse pool of participants, future course organizers could also reach out to local networks and collectives of naturalists of color. Studies have shown that groups that encourage a sense of social belonging among their participants, including among underrepresented minorities, also benefit from active learning environments and promote learning gains [61]. Given that this may be a challenge shared across botanical gardens and nature centers, we posit that nature-based courses, such as this, could be offered in various venues, including community gardens and cooperative farms, to engage a broader demographic. An inherent challenge is that these venues must be able to accommodate complex course logistics, which involve indoor components with slide presentations and stereoscope use, and outdoor components where native bees can be seen in natural habitats.

In conclusion, we documented how active-learning experiences in nature-based learning environments can indeed improve people’s attitudes, deepen their appreciation of natural habitats, and increase their knowledge in science topics, such as biodiversity conservation [40,63,64,65]. The “Native Bees of Texas” course is an example of the educational opportunities that academic organizations associated with botanical gardens can offer. The native landscapes, educational facilities, and conservation ethos in the Lady Bird Johnson Wildflower Center offer a truly unique setting for this type of collaboration. This course also underscores the importance of interdisciplinary collaboration among natural and social scientists for the development of science outreach courses. Public engagement initiatives, calibrated to their communities by evaluation results, can have a strong positive impact on local conservation projects and regional conservation policies. Moreover, these initiatives will be more effective in their efforts to educate the public about drivers of global change if they are implemented across a diversity of landscapes and human communities.

## Figures and Tables

**Figure 1 insects-12-00702-f001:**
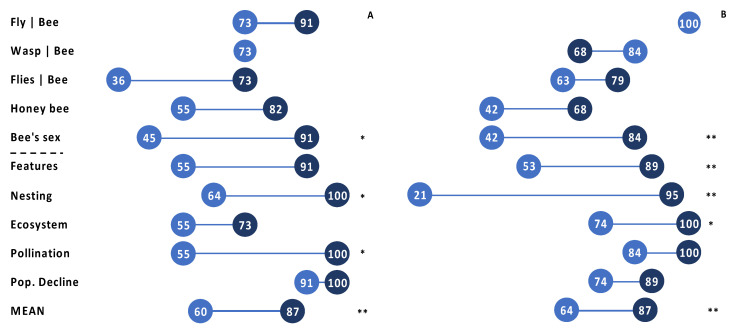
Performance-based Knowledge Gains-Test. Mean proportion (%) of correct pre- and post-course test scores per question across all participants in (**A**) 2018 and (**B**) 2019. The dotted line separates the first five questions focused on Identification Skills from the remaining five questions focused on Ecological Knowledge. * *p* < 0.05, ** *p* < 0.01 indicates statistically significant differences between pre (light) and post (dark) scores (see Appendix A).

**Figure 2 insects-12-00702-f002:**
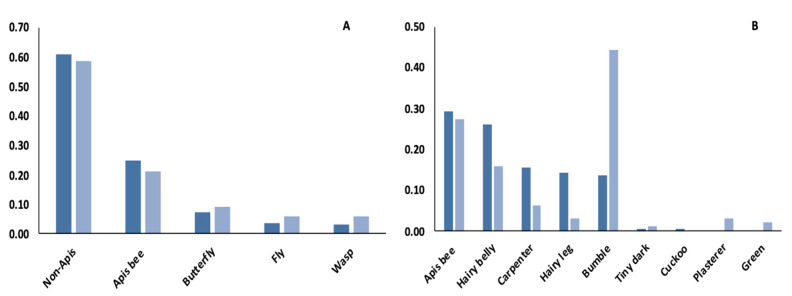
Performance-based Knowledge Gains-Insect Survey. Mean proportion of flower-visiting insects observed by experts (dark) and students (light) during a post-course, paired, insect survey at the Lady Bird Johnson Wildflower Center. Proportional abundance of (**A**) flower-visiting insect groups and (**B**) bee groups (see Appendix A).

**Figure 3 insects-12-00702-f003:**
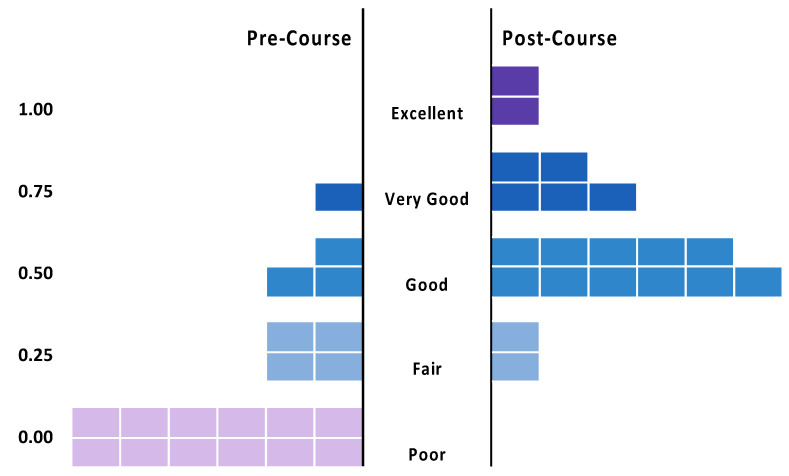
General Perceived Knowledge Gains (2019) Proportion of participants’ retrospective ratings of their general knowledge levels of native bees on a 5-point scale (from light to dark, 1—poor, 2—fair, 3—good, 4—very good, 5—excellent) based on the statement, “I would rate my knowledge of native bees as...”. This translated to a significant increase between the pre-course mean of 1.65 (+/−0.21) and the post-course mean of 3.35 (+/−0.18) (*p* < 0.000) (see Appendix A). Participants are represented by rectangles on the *x*-axis.

**Figure 4 insects-12-00702-f004:**
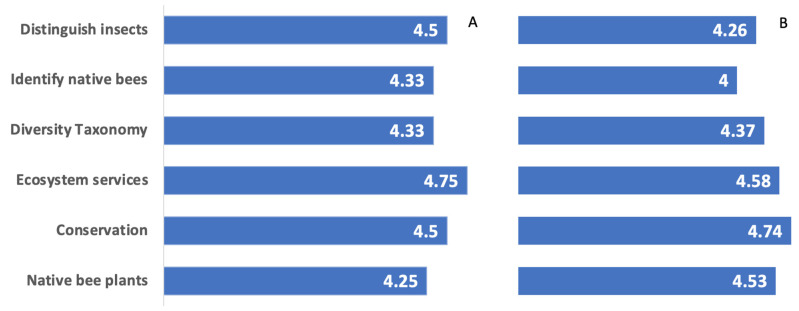
Specific Perceived Knowledge Gains Mean ratings per topic across all participants in (**A**) 2018 and (**B**) 2019 on a 5-point scale (1—strongly disagree, 2—disagree, 3—neutral, 4—agree, 5—strongly agree), where participants rated each of the following 6 specific topics (learning objectives) based on the statement, “As a result of the course, I’m better able to…” (see Appendix A).

## Data Availability

The data presented in this study are openly available in the Texas Data Repository at the following link https://doi.org/10.18738/T8/JTX0TK (accessed on 1 June 2021).

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
