# Peer review of "The Native Bees of Texas: Evaluating the Benefits of a Public Engagement Course"

_insects, 2021, doi:10.3390/insects12080702_

Round 1

Reviewer 1 Report

The manuscript “The Native Bees of Texas: Evaluating the benefits of a public engagement course” presents an interesting way to approach the general public to biodiversity knowledge and raise awareness of conservation topics. I have found it inspiring. Overall, the MS is easy to read and well-built.

One concern I have is related to the structure of the Introduction section. It begins with a paragraph on bee diversity and impediment gap but the subject here is not bee diversity, instead Citizen Science. I think the third paragraph should move up, with a  different beginning, and then the first and second paragraphs. Part of what is reported in the Discussion section (lines 610-626) would do a good job convincing people that your approach is necessary.

Considering the course proposed, in Methods, lines 166-169, it would be very valuable for the readers to explain the relationship between life-history traits and pollination function. It will link better to what is proposed in Appendix S3, Bee ID Guide. It may also help other people to organize similar initiatives.

I was unable to evaluate the methods employed in the topic are up to date “Course evaluation”. They are referenced and seem sound. I also understood the taking this approach is innovative.

In the Discussion section: avoid reporting the results (see for example Lines 456-459; 513-529, 532-534). I suggest synthesizing to the general gains, only addressing specif cases when needed.

Minor comments.

Line 81 – reference 78 before 28 to 77. Renumber-

Lines 195-196: I suggest adding why you did that, perhaps “In the second lab, to distinguish native bees from invasive ones, participants compared honey bees (non-native species) to native bees (Apoidea superfamily)

Line 200 -what does “lecty” stand for?

Inconsistencies: There are some citations reported as a number and others as author-date (especially in the Discussion section)

Appendix S8 – italics missing in scientific names

Author Response

Dear reviewer, 

Re: "The Native Bees of Texas: Evaluating the benefits of a public engagement course" (Insects 1263679). Thank you for your valuable and helpful comments to improve the manuscript, which I will incorporate. I have a question regarding your suggestion: "Part of what is reported in the Discussion section (lines 610-626) would do a good job convincing people that your approach is necessary." Do you mean that this discussion section could be used in the introduction to bolster the argument that this approach is necessary?

Reviewer 2 Report

Dear Authors:

This is a very good study and it should be of interest to educators and researchers in natural science fields, including entomology. My only concern is with the length of the Discussion section. I think it should be shortened somewhat, by removing the text that represents "results" rather than "discussion of results."

Author Response

Greetings, 

I would like to submit my revised manuscript. Should I attach it here?

Laurel Trevino

Reviewer 3 Report

This MS reports a citizen education programme on native bees, it is of conservation significance and worth publication. The MS is generally well presented, data and figures well arranged. I recommendation is minor revision, mainly about the style of discussion and some literature citing and formatting.

Please see the attached PDF with comments.

Author Response

Dear reviewer, 

Thank you for your helpful suggestions to improve the manuscript, which I am incorporating at this time.